# GLP-1 Receptor Agonists in Diabetic Kidney Disease: From Physiology to Clinical Outcomes

**DOI:** 10.3390/jcm10173955

**Published:** 2021-08-31

**Authors:** Alba Rojano Toimil, Andreea Ciudin

**Affiliations:** 1Endocrinology and Nutrition Department, Hospital Universitari Vall Hebron, 08035 Barcelona, Spain; arojano@vhebron.net; 2Institut de Recerca Vall d’Hebron, Universitat Autònoma de Barcelona (VHIR-UAB), 08916 Barcelona, Spain; 3CIBER de Diabetes y Enfermedades Metabólicas Asociadas, Instituto de Salud Carlos III, 28029 Madrid, Spain

**Keywords:** diabetic kidney disease, GLP-1 receptor agonists, diabetic kidney disease

## Abstract

Diabetic kidney disease (DKD) is one of the most common complications in type 2 diabetes mellitus (T2D) and a major cause of morbidity and mortality in diabetes. Despite the widespread use of nephroprotective treatment of T2D, the incidence of DKD is increasing, and it is expected to become the fifth cause of death worldwide within 20 years. Previous studies have demonstrated that GLP-1 receptor agonists (GLP-1 RA) have improved macrovascular and microvascular outcomes independent of glycemic differences, including DKD. GLP-1Ras’ improvement on kidney physiology is mediated by natriuresis, reduction in hyperfiltration and renin-angiotensin-aldosterone system (RAAS) activity and anti-inflammatory properties. These findings translate into improved clinical outcomes such as an enhanced urine albumin-to-creatinine ratio (UACR) and a reduction in renal impairment and the need for renal replacement therapies (RRT). In this article, we review the role of GLP-1RAs on the mechanisms and effect in DKD and their clinical efficacy.

## 1. Introduction

In the past 20 years, the management of type 2 diabetes (T2D) and its complications has improved substantially. Control of cardiovascular risk factors, such as dyslipidemia, obesity, smoking and changes in other modifiable factors based on strict blood pressure control, use of RAAS inhibitors and adequate glycemic treatment, has resulted in improvement in the incidence in T2D related myocardial infarction, stroke, amputations and mortality [1,2,3]. However, diabetic kidney disease (DKD) remains the leading cause of chronic kidney disease (CKD), kidney failure (KF) and need for renal transplantation (RT) [4,5]. A study performed in a primary care setting showed that 27.9% of patients with T2D presented some degree of CKD, and the prevalence of patients with a urinary albumin to creatinine ratio (UACR) ≥ 30 mg/g was 15.4% [6]. In the KDIGO study, the percentage of T2D patients with CKD defined as either a glomerular filtration rate (GFR) below 60 mL/min/1.73 m^2^ or UACR ≥ 30 mg/g was 43.1% in those patients aged ≥65 years [7]. All this evidence further shows the importance of optimizing DKD management.

Current standards of diabetes care recommend metformin as initial therapy, after life style modification in patients with T2D. Recently, the guidelines have included in the algorithm for choosing the second line of treatment the criteria of an increased cardiovascular risk, cardiovascular disease or renal complications [8]. In the past decade, two new classes of antihyperglycemics drugs have been introduced in the clinical practice (sodium-glucose cotransporter-2 inhibitors-SGLT2 and GLP1-receptor agonists-GLP-1RAs) that showed a significant decrease in cardiovascular risk and the progression of DKD [9]. At present, there is significant data regarding the role of SGLT-2 inhibitors on DKD (clinical outcomes and mechanisms) [10,11]. Nevertheless, the role of GLP-1-RAs was not so well represented, despite the significant results from different studies.

The GLP1RAs are included in the incretin-based therapies as well as dipeptidyl peptidase 4 (DPP-4) inhibitors. GLP1 is a peptide secreted during ingestion and its mean effect is reflected on glucose metabolism by insulinotropic signals mediated by GLP1 receptors, which are highly expressed on ß-pancreatic cells [12]. This hormone potentiates the insulin secretion and contributes to glucose homeostasis through a wide range of physiological actions not only on pancreatic cells but also in other systems, such as the central nervous system (CNS) through the regulation of homeostatic feeding and the gastrointestinal tract by slowing gastric emptying [13,14]. The main limitation of the native GLP-1 is the short half-life (3-5 min) due to rapid degradation by the DPP-IV enzyme.

The first drug from the GLP-1RAs’ class was exenatide, which was approved by the US Food and Drug Administration (FDA) in April 2005 for the treatment of T2D. It consists in a 39-aminoacid peptide made by a substitution of an Ala8Gly of the exendin-4 hormone found in the saliva of the Gila monster. This modified chemical structure confers resistance to the degradation of DPP-4 [15]. From that moment, several GLP-1 peptides were modified to confer resistance to DPPIV, limit renal clearance and delay subcutaneous tissue absorption and therefore increase half-life. On these bases, we can classify GLP1-RAs in short-acting or prandial, such as exenatide and liraglutide, with a duration up to 2–4 h, and long acting GLP1-RAs, that have half-lives of up to a week such as albiglutide, dulaglutide and semaglutide [14,16]. Diverse effects in terms of improved weight loss [17,18] and a reduction in glycated hemoglobin (HbA1c) [19] have been demonstrated.

The lack of trials in DKD has limited the experience of pharmacokinetic and clinical data with GLP-1AR in these group of patients with T2D. Therefore, caution or discontinuation was recommended when renal function is severely impaired [20]. However, in recent trials with GLP1-Ras’ liraglutide and semaglutide that included patients with a GFR up to 15 mL/min/1.73 m^2^, post-hoc analysis demonstrated the safety in patients with CKD [21,22,23], underlying a potential renal protection with this class of drugs.

In this review, we focus on the role of GLP-1 (and GLP-1-RAs) in the physiology of the gut-renal axis, the renoprotective mechanisms of this incretin therapies and the renal outcomes obtained in the most recognized clinical studies using GLP1-RAs in patients with T2D.

## 2. The Association of GLP-1 and Gut-Renal Axis

As mentioned above, GLP-1 and GIP are two incretin hormones responsible for the reduction in blood glucose levels in response to nutrient ingestion. In fact, humans submitted to an oral glucose load showed a much greater increase in plasma insulin levels than those infused with intravenous glucose administration [12]. This phenomenon is known as the incretin effect, and it is estimated to account for approximately 50–70% of the overall insulin secretory response after nutrient ingestion [24].

Nevertheless, not only a gut-pancreas connection has been demonstrated but also a contribution to the physiological control of water and electrolyte balance upon meal ingestion. The regulation of the gut-renal axis is mediated by multiple pleiotropic actions in different locations such as the central nervous system, adjusting thirst, intestinal co-transporters to control fluid and electrolyte absorption and secretion and also on the kidney, by the stimulation of renal tubular excretion and/or reabsorption of fluid and electrolytes [25,26]. As the effect previously exposed with oral glucose administration, GLP1 release by oral sodium load can stimulate more rapidly the tubular excretion by the kidney than given intravenously, independent of changes in the levels of aldosterone and atrial natriuretic peptide on plasma [27,28]. The same changes have been observed on potassium and phosphate metabolism [29].

The role of GLP1 is not only limited on tubular effects but also on changes on hemodynamics. For example, a postprandial hyperfiltration induced after ingestion of a high protein meal by an increase in renal blood flow was identified. This mechanism contributes in part of sodium and solutes homeostasis after ingestion by increasing the pressure and vasodilation of the afferent arteriole through nitric oxide [30]. From a pathophysiological point of view, an impaired gut-renal axis implies a reduction in urinary secretion and, therefore, a salt-sensitive hypertension. The glomerular hyperfiltration classically has been related to T2D and the CKD progression. Nevertheless, the postprandial hyperfiltration induced by GLP1 is minimal and has no clinical impact [31].

## 3. Renoprotective Mechanisms of GLP1-RAs in T2D

There are several experimental studies that evaluate the effects of GLP1 and GLP1-RAs on the renal metabolism and shed light on the main mechanisms responsible for changes in eGFR, a reduction in albuminuria and other renal outcomes seen in clinical studies. In this section, we will explain in detail the main mechanisms underlying the attenuation of DKD by GLP1-RAs.

### 3.1. Glucose Lowering

The main renoprotective effect of incretin-based therapies such GLP1-RAs is mediated by regulation of the glucose metabolism. Indeed, the most known effect of the GLP1 gut derived hormone is the reduction in hyperglycemia in T2D which is severely impaired or lost on those patients [32,33]. Its mechanism is based on increasing the insulin secretion and synthesis in pancreatic islet cells and a decrease in glucagon secretion and β-cell apoptosis [34,35]. In fact, it has been published that GLP-1RAs reduce ${\mathrm{HbA}}_{1c}$ levels by ~1.0% compared with a placebo [19]. Other actions that contribute to glucose homeostasis include the diminution of gastric emptying and small intestine peristalsis and suppression of endogenous glucose production [36,37]. Another fact that supports this proposition is that the reduction in new-onset albuminuria in clinical studies with GLP-1RA is accompanied by effective glucose lowering [38].

### 3.2. Oxidative Stress and Inflammation

Various reports have shown that T2D is associated with chronic low-grade inflammation, which is linked with oxidative stress, proliferation and fibrosis that affect kidney function and morphology [39]. Experimental studies have reported that GLP1-RAs inhibit inflammatory signaling pathways of DKD, such as nicotinamide adenine dinucleotide phosphate (NADPH) oxidase by activating protein kinase A (PKA) and the production of cyclic adenosine monophosphate (cAMP), which is paralleled by reductions in albuminuria and improved histological features of DKD [40]. For example, exendin-4 was shown to inhibit proliferation and fibrosis in human mesangial cells by stimulation of cAMP and PKA [41].

Another main molecular target implicated in oxidative stress and progression to DKD is the nuclear factor kappa-light-chain-enhancer of activated B cells (NF-kB). The activation of this pro-inflammatory protein complex is induced by hyperglycemia and attenuates the therapeutic effects of the GLP-1R agonist by its downregulation [42]. The beneficial effects of GLP1-R signaling might also be mediated by increasing eNOS endothelial levels and inhibiting the expression of TNF-alfa in podocytes, both mediated by NF-kB downregulation [32,43]. Similarly, liraglutide has been shown to reduce the structural damage of podocytes in a model of glomerulopathy related to obesity [32]. Additionally, in rodent models treated with liraglutide, this GLP-1RA also modulates other pathways involved DKD, such as JAK/STAT and MAPK pathways present in kidney endothelial cells in rat CKD models [19,33].

### 3.3. Natriuresis—Tubular Effect

GLP-1 has been previously demonstrated to induce renal sodium excretion and increase urine flow in experimental and clinical studies in healthy and T2D individuals [44,45,46]. This effect is carried out by GLP1-RA inhibition of sodium-hydrogen exchanger 3 (NHE3), a channel localized in the apical membrane of the epithelial cells of the proximal tubule on the nephron. This process is mediated by PKA activation through cAMP generation and, finally, NHE3 phosphorylation [47,48]. Additionally, GLP-1RA has an indirect effect also, by influencing the RAAS by reducing renin and angiotensin II circulating levels and action. [49,50]. Furthermore, GLP1-RAs were described to play a role in natriuresis and diuresis, by modulating the tubular ionic exchange of potassium, chloride and calcium [49,51,52].

### 3.4. Endothelial Function—Glomerular Effect

It is well known that the mechanism of hyperfiltration is a prime disruptor in DKD, especially when it is accompanied by albuminuria and renal function decline, which is observed in advance stages of the disease [53]. The GLP1-RAs’ effect on glomerular hemodynamics is still controversial and remains to be elucidated.

In rodent models, GLP1-RA mediated the increase in endothelial nitric oxide synthase (eNOS) activity and expression, and nitric oxide (NO) production was described. This NO increase produces a direct vasodilation effect mainly on preglomerular arterioles that increases the glomerular filtration rate (GFR) [54,55]. Additionally, the increase in GFR and effective renal plasma flow (ERPF) induced by NO-dependent glomerular afferent arteriole vasodilation has been reported in numerous preclinical studies during short-term interventions with GLP-1 and GLP-1RAs [50,56,57]. It should be mentioned that all these models have been performed in rodents without diabetes, and the dose administration of GLP-1 and GLP1-RAs exceeds human therapeutic concentrations. In humans, it has been reported in one clinical study that exenatide infusion in ten healthy overweight men increased inulin-measured GFR and ERPF [58].

On the other hand, other clinical and animal studies suggest that GLP1-RAs’ therapies reduce glomerular hyperfiltration and provide renoprotection in DKD. In experimental studies in diabetic rodents, 4–8 weeks of administration of liraglutide and linagliptin significantly reduced glomerular hyperfiltration [59]. In clinical studies, it has been observed that single dose GLP-1 infusion reduced GFR measurements in 16 subjects with obesity that presented hyperfiltration [37]. Additionally, liraglutide produced a decrease in GFR and albuminuria in patients with T2D and normal filtration [59].

These discrepancies on GLP1-RAs’ effects on renal hemodynamics may be explained by differences in the characteristics of the studied population, the dose administrated and the posology or differences in study designs.

### 3.5. Blood Pressure

GLP1RAs treatment points to a clinically relevant lowering effect on blood pressure. These results can be partially explained by indirect effects on weight loss, an increase in natriuresis and the regulation of RAAS [60,61,62]. Moreover, independent effects on NO dependent vasorelaxation as well as changes in endothelial cell function could be involved. A single-blind randomized crossover study on 12 patients with T2D with stable coronary artery disease that underwent intravenous infusion of human recombinant GLP-1 described a significant increase in the brachial artery diameter. This effect was proven to be mediated by the GLP-1 via expression of the GLP1-receptor in endothelial cells (western blotting of cell lysates) [63]. Similar effects have been observed in the femoral artery after exendin infusion in rats [64]. In a meta-analysis of 60 randomized control trials, blood pressure was only significantly reduced with liraglutide and albiglutide compared with a placebo; a non-statistical effect was observed with exenatide and dulaglutide [61]. Evidence to link blood pressure reduction to GLP-1R signaling and mechanisms is scarce and only partially understood.

### 3.6. Dyslipidemia

GLP1-RAs have a main glucose lowering effect by stimulating β-cell pancreatic secretion and inhibiting glucagon production. Strict control of dyslipidemia was shown to have a beneficial effect on DKD [62]. This metabolic response implicates changes in the lipid profile resulting in lower levels of triglyceride and low-density lipoproteins, as has been broadly described in literature [65]. Nonetheless, in clinical practice, these changes were marginal in GLP-1 therapies compared to a placebo [66], and the role in DKD remains to be elucidated.

### 3.7. Body Weight

Abdominal obesity is related with T2D and the associated chronic complications such as DKD [67]. Several studies have shown that being overweight is an independent risk factor for CKD and an increase in visceral fat plays the main role in its pathogenesis. In fact, it is well known that the mechanism of renal damage in obesity is very similar to T2DM, with initial augmentation in eGFR and intraglomerular pression and microalbuminuria that culminate in proteinuria, nodular glomerulosclerosis and tubulointerstitial injury [68,69,70].

Other conventional glucose-lowering treatments such as insulin or sulphonylureas increases body weight [71]; however, it is well known that GLP1-RAs’ treatment in monotherapy or adding to another conventional treatment can lead to a statistically significant weight loss and reduction in the abdominal perimeter [15]. These findings were reported by some randomized clinical trials [72,73,74] and meta-analyses [75,76]. It is important to note that the predominant effect of these incretin-based therapies is in the reduction of trunk and visceral fat, rather than in lean tissue mass [15,77].

In fact, semaglutide shines as a weight-loss therapy, reaching a decline of 12.4% compared with a placebo in 68 weeks (95% CI, −13.4 to −11.5; *p* < 0.001). This means that, in absolute terms, semaglutide obtained a reduction in 15.3 kg with respect to the baseline (estimated treatment difference, −12.7 kg; 95% CI, −13.7 to −11.7) [18]. Other GLP1RAs’ treatment such as liraglutide or exenatide showed a modest decrease in body weight of −2.51 kg (95% CI, −3.33 to −1.69; *p* < 0.001) and −1.69 kg (95% CI, −2.09 to −1.29; *p* < 0.001), respectively [78,79].

## 4. GLP1-RAs and Renal Outcomes in Clinical Trials

A literature search was conducted in the following databases for the period through 1 January 2005 to 1 April 2021: Medline, Elsevier, Embase and Scopus. Only randomized, double-blind, placebo-controlled trials published that include renal results as primary or secondary outcomes were considered. Ongoing trials were identified using the US National Institutes Health Clinical Trials Registry (www.clinicaltrials.gov accessed on 25 August 2021). It is important to remark that the data provided about GLP1RAs’ effect on the kidney are obtained of studies that were designed to aim cardiovascular events, so renal effects were obtained as secondary outcomes—Table 1 (the reduction in renal outcomes was signaled by “↓”).

### 4.1. Renal Outcomes with Liraglutide in the LEADER Trial

In this study, 9340 participants with T2D and high cardiovascular risk were assigned to liraglutide 1.8 mg subcutaneous injection vs. placebo and were followed 3.84 years. The renal outcomes studied included new-onset persistent macroalbuminuria, persistent doubling of serum creatinine, KF and death due to renal disease. All the renal events were grouped on first statistical analysis, and a reduction in its incidence was observed as HR 0.78 (95% CI: 0.67–0.92), and the main contribution of this effect was the reduction in new onset macroalbuminuria HR 0.74 (95% CI 0.60–0.91, *p* = 0.004). No differences in the doubling of the serum creatinine, initiation of RRT or renal death were observed. On *post hoc* analysis, liraglutide showed a reduction in the risk of major adverse CV events and all-cause mortality in comparison to a placebo in patients with DCKD (eGFR < 60 mL/min/1.73 m^2^ and UACR > 30 mg/g) [80,81].

### 4.2. Renal Outcomes with Semaglutide in the SUSTAIN-6 Trial

Semaglutide subcutaneous treatment showed a reduction in the incidence of new or worsening nephropathy vs. a placebo in a SUSTAIN-6 trial (HR 0.64 (95%. CI 0.46–0.88, *p* = 0.05)); the main effect was driven by new onset macroalbuminuria. Concretely, UACR ratios at the end of treatment with respect to baseline were 0.973 with semaglutide 0.5 mg, 0.858 with semaglutide 1.0 mg and 1.302 with the placebo. Other outcomes studied were persistent doubling of the serum creatinine level, creatinine clearance <45 mL/min/1.73 m^2^ and need for RRT. No differences were observed in KF or renal death. The baseline characteristics of the patients were T2D patients with established cardiovascular or chronic kidney disease; 25.2% had an eGFR moderately decreased (30–59 mL/min/1.73 m^2^) and 2.9% severely decreased (<30 mL/min/1.73 m^2^), and 12.7% had macroalbuminuria at screening [19].

SUSTAIN 1–5 and 7 post-hoc analyses showed a 30% reduction in albuminuria and also a regression to micro- or normoalbuminuria was observed in all degrees of albuminuria. Specifically, there was reduction in UACR ratios of 0.74 (95% CI 0.64 to 0.85) with semaglutide 0.5 mg and 0.68 (95% CI 0.59 to 0.78) for semaglutide 1.0 mg versus placebo. A decrease in eGFR were observed at the start of the study in patients with normal kidney function (and in those with mild kidney impairment with semaglutide 1.0 mg in SUSTAIN 6), but no differences were found at the end of the study between semaglutide and the placebo. No increase in the risk of kidney adverse events with semaglutide was seen [19].

### 4.3. Renal Outcomes with Semaglutide in the PIONEER-6 Trial

The main objective of this trial was the evaluation of cardiovascular safety of oral semaglutide. At baseline, 31283 participants with high cardiovascular risk, ≥50 years of age and a cardiovascular and/or chronic disease established were included. It is important to note that 26.9% of the patients had an eGFR < 60 mL/min/1.73 m^2^. Subjects were followed up for 15.9 months. In the results, no statistical differences were observed in the renal outcomes studied (eGFR decline and renal death). No renal outcomes were analyzed in PIONEER-5, but safety was demonstrated for individuals with a moderate eGFR decrease (30–59 mL/min/1.73 m^2^) [82,83].

### 4.4. Renal Outcomes with Semaglutide in the FLOW Trial

The FLOW trial is the first and ongoing study with renal outcome as the primary objective, defined as persistent EGFR decline (≥50% from the start point), reaching KF or renal death from CV or kidney disease. The enrollment of 3508 participants is estimated in a randomized, quadruple-blind, placebo-controlled trial that evaluates the effect of semaglutide on the progression of renal disease in patients with CKD and T2D. The inclusion criteria are eGFR ≥ 50 mL/min/1.73 m^2^ and ≤75 mL/min/1.73 m^2^, and the UACR between 300 and 5000 mg/g or eGFR ≥ 25 mL/min/1.73 m^2^ and ≤50 mL/min/1.73 m^2^ and the UACR between 100 and 5000 mg/g. The dose administrated will be titrated from 0.25 mg to 1 mg, and the estimated finish date will be in August 2024 [84].

### 4.5. Renal Outcomes with Dulaglutide in the AWARD-7 Trial

Renal outcomes were assessed in AWARD-7: this study included 577 patients with T2D and moderate-to-severe CKD treated with insulin and the maximum dose of an angiotensin-converting enzyme inhibitor or an angiotensin receptor blocker. The response to once-weekly injectable dulaglutide 1.5 mg, 0.75 mg, versus daily insulin glargine was evaluated. The primary outcome was HbA1c at 26 weeks. Renal outcomes observed were only secondary objectives, as the estimated glomerular filtration rate (eGFR) and urine albumin-to-creatinine ratio (UACR). The effects of dulaglutide on UACR reduction were not statistically significantly different from that of insulin glargine (22.5% (95% CI −35.1 to −7.5)) with dulaglutide 1.5 mg (20.1% (95% CI −33.1 to −4.6)) with dulaglutide 0.75 mg (95% CI 13.0% (−27.1 to 3.9) with insulin glargine). At 52 weeks, eGFR was higher with dulaglutide 1.5 mg (34.0 mL/min/1.73 m^2^; *p* = 0.005 vs. insulin glargine) and dulaglutide 0.75 mg (33.8 mL/min/1.73 m^2^; *p* = 0.009 vs. insulin glargine) than with insulin glargine (31.3 mL/min/1.73 m^2^). Dulaglutide treatment was associated with a significantly smaller decline in eGFR compared with insulin glargine over 52 weeks. The association between dulaglutide treatment and reduced eGFR decline was most evident in participants with macroalbuminuria [85].

### 4.6. Renal Outcomes with Dulaglutide in the REWIND Trial

In this multicenter, randomized, double-blind and placebo-controlled trial were included 371 participants with at least 50 years of age, with T2D with CV risk factors or a previous CV event. A weekly subcutaneous injection of dulaglutide (1.5 mg) was compared with a placebo. The primary outcome on this study was the composite of the following events: non-fatal myocardial infarction, non-fatal stroke or death from cardiovascular causes (including unknown causes). On a follow-up of 5.4 years, the renal outcome (defined as the junction of new macroalbuminuria, sustained decline in eGFR of 30% or more and need of RRT) occurred in 17.1% of participants sorted in the dulaglutide treatment branch and in 19.6% in the placebo group (HR 0.85, 95% CI 0.77–0.93; *p* = 0.0004). New onset macroalbuminuria was the only renal outcome to be statistically significant when studied separately (HR 0.77 95% CI 0.68–0.87; *p* = 0.0001); no differences were observed in the sustained decline in eGFR (HRs of 0.89 (0.78–1.01; *p* = 0.066)) and chronic renal replacement therapy (0.75 (0.39–1.44; *p* = 0.39)) [86].

### 4.7. Renal Outcomes with Exenatide in the EXSCEL Trial

The EXSCEL 2017 trial assessed the cardiovascular safety of exenatide in 14.752 patients of whom 73.1% had previous cardiovascular disease, and were followed for a median of 3.2 years. The primary outcome composite of the first occurrence of death from cardiovascular causes, nonfatal myocardial infarction or nonfatal stroke occurred in 11.4% in the exenatide group and in 12.2% in the placebo group (HR, 0.1; 95% CI, 0.83 to 1.00). Exenatide therapy was not associated with significant differences in eGFR decline, RRT or renal death. However, based on the data obtained, macroalbuminuria occurred less in the exenatide group (2.2%) compared to the placebo (2.5%) (HR 0.87 (95% CI: 0.70–1.07)) [87]. In post hoc analysis, exenatide did not show statistical differences in any renal outcome studied; neither creatinine clearance, eGFR decline nor new onset progression of albuminuria were observed [88].

### 4.8. Renal Outcomes with Albiglutide in the HARMONY Trial

In HARMONY, the incidence of major adverse cardiovascular events in patients aged 40 years and older with T2D and CV disease occurred in 338 of 4731 (7%) patients that received a subcutaneous injection of albiglutide and in 428 (9%) of 4732 patients in the placebo group (hazard ratio 0.78, 95% CI 0.68–0.90). They only examined decline in the estimated glomerular filtration rate as a parameter of kidney outcome, and there were not significant differences in the treatment group versus the placebo. No other renal outcomes were studies [89].

### 4.9. Renal Outcomes with Lixisenatide in the ELIXA Trial

Patients with T2D and a recent acute coronary event (myocardial infarction or unstable angina) were analyzed. This included 6068 patients in a double-blinded placebo-controlled randomized to assess differences in the incidence of cardiovascular death, myocardial infarction, stroke or hospitalization for an unstable angina in individuals treated with a once-daily subcutaneous injection of lixisenatide versus a placebo. In the baseline characteristics, there was a preexisting eGFR decline (<60 mL/min/1.73 m^2^) in 25% of the participants included in the lixisenatide group and 22% treated with the placebo. In ELIXA, a prespecified analysis of the percentage change in UACR from baseline was employed, and showed a modest difference over the placebo (24% vs. 34%, *p* = 0.004) [70]. In a post hoc model, changes in the eGFR rate and the UACR adjusted at baseline for albuminuria status were assessed. No statistically significant differences were seen in eGFR decline for lixisenatide treatment versus the placebo overall or adjusted at baseline for albuminuria status. However, in this analysis was obtained a decrease of risk in the first macroalbuminuria event in patients without macroalbuminuria at the start point (HR 0.77; 95% CI: 0.62, 0.96; *p* = 0.0174) and a lower UACR change rate in patients treated with lixisenatide with micro- (−21.10%, −42.25 to 0.04; *p* = 0.0502) or macroalbuminuria (−39.18%, −68.53 to −9.84, *p* = 0.0070). All these results were adjusted to HbA1c changes, demonstrating a reduced UACR progression beyond glycemic control [90].

## 5. Conclusions and New Perspectives

DKD remains the leading cause of chronic kidney disease, kidney failure and need for RRT [4,5]. The classical treatment of this diabetic complication was the control of cardiovascular risk factors and optimization of blood pressure and glycemia [1,2]. However, new treatments are emerging, and GLP1-RAs seem to be an optimal option in patients with T2D and CVD. This hypothesis has been demonstrated by multiple clinical trials based on large cohorts of populations [80,81,82,83,84,85,86,87,88,89]. However, renal outcomes always were analyzed as secondary outcomes.

There exists several pre-clinical studies tested on animals that demonstrate the beneficial effects of GLP1-RAs on kidney function. In fact, there is evidence of a direct GLP-1 effect on glomerular homeostasis and tubular ion excretion, mechanisms implicated on hyperfiltration and the main histopathological cause of DKD [45,49,51]. Other effects such as endothelial protection mediated by the inhibition of inflammatory pathways as NAPDH oxidase have been objectivized [40,41]. All these mechanisms were independent of glucose lowering achieving.

Due to the evidence previously exposed, the development of new large clinical trials in the T2D population with DKD is urgently needed to analyze the main effect of GLP1-RAs on the kidney as primary objectives. In fact, there are currently ongoing several clinical trials with GLP1-RA with renal outcomes as the main hypothesis: the effect of liraglutide, lixisenatide and semaglutide on renal complications, such as eGFR decrease or albuminuria progression, is being further studied (NCT01847313, NCT02276196, NCT03819153). This new perspective offers the possibility to reduce the incidence of patients with moderated-advanced stages of CKD as well as the need for RRT or kidney transplant, resulting in a decrease in hospitalizations and deaths due to renal causes and, consequently, a reduction in sanitary costs without sacrificing the quality of life of patients with CKD.

## Figures and Tables

**Table 1 jcm-10-03955-t001:** GLP1-RAs and renal outcomes in main clinical trials.

Study	N	Study Design	Baseline Characteristics	Median Follow Up (years)	Renal Outcome Studied	Results
LEADER NCT01179048	9340	DB-RCT	T2D + high CV risk	3.8	New-onset macroalbuminuria, doubling of the serum creatinine level, KF, renal death	↓ Renal outcomes HR 0.78 (95% CI: 0.67–0.92)
SUSTAIN-6 NCT01720446	3297	DB-RCT	T2D + ≥50 years + established CVD/CKD stage III-V T2D + ≥60 years + CV risk factors	2.1	New or worsening of nephropathy (persistent macroalbuminuria, doubling of the serum creatinine level and CCr < 45 mL/min/1.73 m^2^, RRT)	↓ Renal outcomes HR 0.64 (95% CI: 0.46–0.88)
PIONEER-6 NCT02692716	31,283	DB-RCT	T2D + ≥50 years + established CVD/ CKD stage III-V	1.3	Changes in eGFR decline and rate of renal related death	No statistical differences
LOW NCT03819153	3508	QB-RCT	T2D + established CKD stage III-IV	5	Time to first occurrence of persistent eGFR decline (≥50%), reaching KF, death from KD or CV, annual rate of change in eGFR, change in eGFR, relative change in UACR	
AWARD-7 NCT01621178	577	OL-RCT	T2D + established CKD stage III-IV	0.997	Changes in eGFR decline and UACR from baseline	No statistical differences
REWIND NCT01394952	9901	DB-RCT	T2D + previous CV event/CV risk factors	5.4	New onset of macroalbuminuria, sustained eGFR decline (≥30%) or RRT	↓ Renal outcomes HR 0.85 (95% CI: 0.77–0.93)
EXSCEL NCT01144338	14,752	DB-RCT	T2D (70% with previous CV event)	3.2	New-onset macroalbuminuria, 40% eGFR decline, KF, renal death	↓ Renal outcomes HR 0.87 (95% CI: 0.70–1.07)
HARMONY NCT02465515	9463	DB-RCT	T2D + ≥40 years + established CVD	1.6	Changes in eGFR decline	↓ Renal outcomes HR 0.78 (95% CI: 0.68–0.90)
ELIXA NCT01147250	6068	DB-RCT	T2D + recent acute coronary event	2.1	Percent change in UACR and eGFR from baseline	Lower UACR CR (−21.10%, −42.25 to 0.04; *p* = 0.0502 in mAlb); (−39.18%, −68.53 to −9.84, *p* = 0.0070 on MAlb)

## Data Availability

Not applicable.

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
