# Peer review of "GLP-1 Receptor Agonists in Diabetic Kidney Disease: From Physiology to Clinical Outcomes"

_jcm, 2021, doi:10.3390/jcm10173955_

Round 1

Reviewer 1 Report

Overall:

This review evaluates the effects of GLP-1 receptor antagonism in clinical studies focusing on renal outcomes. Primarily the authors identify that there are limited studies to date that sufficiently explore the clinical potential of GLP-1 RA therapy in diabetes associated chronic kidney disease. Thus the review is topical and of general interest to the field.

This review is presented in two main sections followed by a discussion of the overall conjecture. Section 2 briefly describes the results of GLP-1 antagonism on the physiological effects most frequently observed with T2DM. They provide some great insight into the physiological effects of these compounds by briefly reviewing many of the key renoprotective mechanisms. Section 3 presents a summary of 8-9 individual clinical studies with renal outcomes available under GLP1-RA treatment. Overall the review is brief yet raises some insightful points about the research reviewed. However, there are some key details that should be included to help orient the reader to the author’s perspective and improve clarity:

Broad points for consideration:

1. This review would benefit from some additional details to ensure the audience is familiarised with the subject matter. Particularly, a short paragraph should be included that broadly describes the mechanisms of GLP-1 signalling and importance of GLP-1 receptor antagonism. This should include an overview of the current therapeutic strategies and the broad classes of drugs that are in clinical use.

2. In section 2, each subsection describes a handful of studies for each main effect. The section of glucose lowering is particularly short. As this is described as the main renoprotective effect of GLP-1 RA and is thus a significant consideration, this paragraph should discuss the clinical importance of this mechanism in more detail. It may also be beneficial to place this subsection first.

3. Section 3 appears to be structured as a mini review of available clinical studies however it is unclear how these studies were found and included (I.e. was a search criteria used or other methodology?). This should be stated  clearly at the end of the introduction along with a specific description of inclusion criteria.

4. Section 3 would benefit from some structured analysis to compare/contrast the results of the studies rather than summarising the result of each study individually. Otherwise, this section adds very little insight to the field.

Specific points for consideration:

  1. Line 39: Can you suggest a good reference for SGLT-2 inhibitors?
  2. Line 45 to 47: Check phrasing
  3. Line 49: spelling – ‘of’
  4. Line 58: Change “additionally” to “for example” to better identify that exendin-4 is a GLP-1-RA example, following the previous sentence.
  5. Line 61: phrasing is unclear, perhaps: Another main oxidative stress target in DKD…
  6. Line 62-63: Rephrase sentence to better reflect the results of the study in reference 12.
  7. Line 68: specify which molecules/peptides have been used in the studies referenced.
  8. Abbreviation for RAAS should be consistent across the manuscript. (See lines 81 and 114)
  9. Please double check references, as some do not appear to match up with the text (see reference 40, and 42 in line 118-120).
  10. Reference 45 reports the difference between short and long acting compounds and they do report a difference with the long acting compounds.
  11. Some of the results of the studies cited in Table 1 have been missed (eg eGFR reduction by dulaglutide in the AWARD-7 study).
  12. The FLOW Trial is currently missing from Table 1.
  13. Please also include the clinical trial numbers for each study included in Table 1, where available, as in line 136.

Author Response

REVIEWER 1

Comments and Suggestions for Authors

Overall:

This review evaluates the effects of GLP-1 receptor antagonism in clinical studies focusing on renal outcomes. Primarily the authors identify that there are limited studies to date that sufficiently explore the clinical potential of GLP-1 RA therapy in diabetes associated chronic kidney disease. Thus the review is topical and of general interest to the field.

This review is presented in two main sections followed by a discussion of the overall conjecture. Section 2 briefly describes the results of GLP-1 antagonism on the physiological effects most frequently observed with T2DM. They provide some great insight into the physiological effects of these compounds by briefly reviewing many of the key renoprotective mechanisms. Section 3 presents a summary of 8-9 individual clinical studies with renal outcomes available under GLP1-RA treatment. Overall the review is brief yet raises some insightful points about the research reviewed. However, there are some key details that should be included to help orient the reader to the author’s perspective and improve clarity:

Answer: Thank you for your kind comments and the revision process, that have significantly increased the quality of our paper.

Broad points for consideration:

  1. This review would benefit from some additional details to ensure the audience is familiarised with the subject matter. Particularly, a short paragraph should be included that broadly describes the mechanisms of GLP-1 signalling and importance of GLP-1 receptor antagonism. This should include an overview of the current therapeutic strategies and the broad classes of drugs that are in clinical use.

Answer: First of all we assumed that there was a typo error when the reviewer referred to “GLP-1 receptor antagonism” and not “GLP-1 receptor agonism”.  As suggested, we have included a short paragraph regarding the current standards of care in T2D, the different drugs involved, the main incretin therapies and a brief resume of the description of the different GLP-1 that are actually approved by the FDA.

  1. In section 2, each subsection describes a handful of studies for each main effect. The section of glucose lowering is particularly short. As this is described as the main renoprotective effect of GLP-1 RA and is thus a significant consideration, this paragraph should discuss the clinical importance of this mechanism in more detail. It may also be beneficial to place this subsection first.

Answer: We have modified the glucose lowering section, as suggested.

  1. Section 3 appears to be structured as a mini review of available clinical studies however it is unclear how these studies were found and included (I.e. was a search criteria used or other methodology?). This should be stated  clearly at the end of the introduction along with a specific description of inclusion criteria.
  2. Section 3 would benefit from some structured analysis to compare/contrast the results of the studies rather than summarising the result of each study individually. Otherwise, this section adds very little insight to the field.

Answer 3 and 4: In this review we conducted a literature search in the following databases for the period through 1 January 2005 to 1 April 2021: Medline, Elsevier, Embase and Scopus. Only randomized, double-blind, placebo-controlled trials published that include renal results as primary or secondary outcomes were considered. Ongoing trials were identified using the US National Institutes Health Clinical Trials Registry (www.clinicaltrials.gov). Thank you for the recommendation, and this information has been added to the review.

Specific points for consideration:

  1. Line 39: Can you suggest a good reference for SGLT-2 inhibitors? Thank you for the advice, we added reference to the statement.
  2. Line 45 to 47: Check phrasing For better understanding of the audience, we have changed the paragraph.
  3. Line 49: spelling – ‘of’ Thank you, changed.
  4. Line 58: Change “additionally” to “for example” to better identify that exendin-4 is a GLP-1-RA example, following the previous sentence. Thank you, changed.
  5. Line 61: phrasing is unclear, perhaps: Another main oxidative stress target in DKD… For better understanding of the audience, we have changed the paragraph.
  6. Line 62-63: Rephrase sentence to better reflect the results of the study in reference 12. We have written another sentence to consider the results in appropriate way.
  7. Line 68: specify which molecules/peptides have been used in the studies referenced. Thank you, changed.
  8. Abbreviation for RAAS should be consistent across the manuscript. (See lines 81 and 114). We have changed the abbreviation to be consistent across the manuscript.
  9. Please double check references, as some do not appear to match up with the text (see reference 40, and 42 in line 118-120). We have corrected the error of the references, Thank you.
  10. Reference 45 reports the difference between short and long acting compounds and they do report a difference with the long acting compounds. We have changed the error of the references, Thank you.
  11. Some of the results of the studies cited in Table 1 have been missed (eg eGFR reduction by dulaglutide in the AWARD-7 study). Thank you, we have included on the text all the results of the table.
  12. The FLOW Trial is currently missing from Table 1. Initially we aimed to include only the finished studies, but now we have also included FLOW trial as suggested.
  13. Please also include the clinical trial numbers for each study included in Table 1, where available, as in line 136. Thank you, changed.

We hope that we have properly addressed all the issues and you find this reviewed manuscript suitable for publication in the Journal of Clinical Medicine.

Yours sincerely,

Andreea Ciudin MD, PhD

Endocrinology and Nutrition Department
Morbid Obesity Unit Coordinator-EASO acredited Center of Excellence-COM
Hospital Universitari Vall d´Hebron
Universitat Autònoma de BarcelonaPg Vall d´Hebron 119-129, 08035, Barcelona
0034932744736/0034932746591e-mail: aciudin@vhebron.net

Reviewer 2 Report

I had the opportunity to read the manuscript entitled “GLP-1 Receptor Agonists in Diabetic Kidney Disease: from 2 Physiology to Clinical Outcomes” Alba Rojano Toimil 1, Andreea Ciudin, who intent to publish in International Journal of Clinical Medicine.

Authors have chosen a very important and widely worked upon topic for clinical review. They have worked hard and did a great job to compile all the important aspects  the most important aspects for the use of GLP-1AR in patients with Diabetic Kidney Disease (DKD).

From a point of informative medicine, the manuscript is well redacted and focused.

Some minor concerns

1)

In page 1 28 can be see “end-stage renal disease (ESRD)”. I suggest adapt “Nomenclature for Kidney Function and Disease: Executive Summary and Glossary from a Kidney Disease: Improving Global Outcomes (KDIGO) Consensus Conference. They recommend avoid the term “end-stage.”  The term “kidney failure,” which is defined as GFR <15 ml/min per 1.73 m2 or treatment by dialysis, is as comprehensive as “ESRD/ESKD,” without suffering from its limitations.

2)

In the introduction the authors explain “… we focus on the role of GLP-1 (and GLP-1-RAs) in the physiology of 41 the gut-renal axis … (page 1 line 41), but little can be read about the gut-renal axis in the text. Nevertheless, they have included very nice revisions in Bibliography (Muskiet, M. H. A.; Smits, M. M.; Morsink, L. M.; Diamant, M. The Gut-Renal Axis: Do Incretin-Based Agents Confer Reno-201 protection in Diabetes? Nat Rev Nephrol 2014, 10 (2), 88–103. and Yang, J.; Jose, P. A.; Zeng, C. Gastrointestinal–Renal Axis: Role in the Regulation of Blood Pressure. Journal of the American 252 Heart Association 6 (3), e005536.).

I suggest include something more about gut-renal axis because it can facilitate understating why GLP-1RA, depending on the clinical context (acute vs chronic administration, fast-acting or slow-acting analogues, diabetic patients vs healthy individuals, hyperfiltration vs normal or reduced kidney function at the beginning of the study, co-medication with ACE inhibitors, ARBs ...) these drugs can increase or decrease glomerular filtration. In Muskiet et al Nat Rev Nephrol 2014 we can read that the gut has been suggested to directly detect changes in the levels of ingested electrolytes and couple these changes to release of hormones and/or activation of neural pathways that regulate renal tubular and gastrointestinal transport.GLP-1 can be one of gut hormones and peptides implicated in gut-renal axis. The mechanism by which GLP-1 contributes to the increased excretion of sodium and solutes after ingestion is, in part, by increasing blood pressure and vasodilation of the afferent arteriole (through nitric oxide), increasing the renal blood flow and glomerular filtration, producing pressure natriuresis. Even, an impaired gut–renal axis in urinary sodium excretion might contribute to salt-sensitive hypertension. Although the natriuretic effect and the effect on glomerular filtration are adequately explained in the text, it is not related to the gut-renal axis.

3)

In my opinion, one of the most interesting results of AWARD-7 is that at 52 weeks, eGFR was higher with dulaglutide 1∙5 mg (Chronic Kidney Disease Epidemiology Collaboration equation by cystatin C geometric LSM 34∙0 mL/min per 1∙73 m² [SE 0∙7]; p=0∙005 vs insulin glargine) and dulaglutide 0∙75 mg (33∙8 mL/min per 1∙73 m² [0∙7]; p=0∙009 vs insulin glargine) than with insulin glargine (31∙3 mL/min per 1∙73 m²). Dulaglutide treatment was associated with a significantly smaller decline in eGFR compared with insulin glargine over 52 weeks. These data suggest that dulaglutide could have specific therapeutic benefits that might slow progression of moderate-to-severe chronic kidney disease in type 2 diabetes. The association between dulaglutide treatment and reduced eGFR decline was most evident in participants with macroalbuminuria. Is the only study with GLP-1RA in which there was a reduced decline in eGFR. Although dulaglutide treatment was associated with a reduced decline in eGFR, the study was not designed to assess clinical endpoints because of the short 1-year treatment duration. Differences in rates of chronic kidney disease progression or end-stage renal disease will require long-term studies to substantiate that the eGFR benefit of GLP-1RA treatment translates to reduced rates of these events. (Tuttle KR, Lakshmanan MC, Rayner B, Busch RS, Zimmermann AG, Woodward DB, et al. Dulaglutide versus insulin glargine in patients with type 2 diabetes and moderate-to-severe chronic kidney disease (AWARD-7): a multicentre, open-label, randomised trial. Lancet Diabetes Endocrinol [Internet]. 2018 Aug 1 [cited 2018 Oct 13];6(8):605–17)

4)

Another interesting question that in my opinion should be mentioned is that these drugs can be used up to a GFR of 15 mL/min/1.73 m2 (Górriz, J.L.; Soler, M.J.; Navarro-González, J.F.; García-Carro, C.; Puchades, M.J.; D’Marco, L.; Martínez Castelao, A.; Fernández- Fernández, B.; Ortiz, A.; Górriz-Zambrano, C.; et al. GLP-1 Receptor Agonists and Diabetic Kidney Disease: A Call of Attention to Nephrologists. J. Clin. Med. 2020, 9, 947). This fact implies the importance of this type of hypoglycemic agents for the nephrological community and DKD patients.

Author Response

REVIEWER 2

Comments and Suggestions for Authors

I had the opportunity to read the manuscript entitled “GLP-1 Receptor Agonists in Diabetic Kidney Disease: from 2 Physiology to Clinical Outcomes” Alba Rojano Toimil 1, Andreea Ciudin, who intent to publish in International Journal of Clinical Medicine.

Authors have chosen a very important and widely worked upon topic for clinical review. They have worked hard and did a great job to compile all the important aspects  the most important aspects for the use of GLP-1AR in patients with Diabetic Kidney Disease (DKD).

From a point of informative medicine, the manuscript is well redacted and focused.

Answer: Thank you for your kind comments and the revision process, that have significantly increased the quality of our paper.

Some minor concerns

 1)In page 1 28 can be see “end-stage renal disease (ESRD)”. I suggest adapt “Nomenclature for Kidney Function and Disease: Executive Summary and Glossary from a Kidney Disease: Improving Global Outcomes (KDIGO) Consensus Conference. They recommend avoid the term “end-stage.”  The term “kidney failure,” which is defined as GFR <15 ml/min per 1.73 m2 or treatment by dialysis, is as comprehensive as “ESRD/ESKD,” without suffering from its limitations.

Answer: Thank you for your suggestion, we have changed the nomenclature as the recommendations of KDIGO Consensus Conference.

2)In the introduction the authors explain “… we focus on the role of GLP-1 (and GLP-1-RAs) in the physiology of 41 the gut-renal axis … (page 1 line 41), but little can be read about the gut-renal axis in the text. Nevertheless, they have included very nice revisions in Bibliography (Muskiet, M. H. A.; Smits, M. M.; Morsink, L. M.; Diamant, M. The Gut-Renal Axis: Do Incretin-Based Agents Confer Reno-201 protection in Diabetes? Nat Rev Nephrol 2014, 10 (2), 88–103. and Yang, J.; Jose, P. A.; Zeng, C. Gastrointestinal–Renal Axis: Role in the Regulation of Blood Pressure. Journal of the American 252 Heart Association 6 (3), e005536.).

I suggest include something more about gut-renal axis because it can facilitate understating why GLP-1RA, depending on the clinical context (acute vs chronic administration, fast-acting or slow-acting analogues, diabetic patients vs healthy individuals, hyperfiltration vs normal or reduced kidney function at the beginning of the study, co-medication with ACE inhibitors, ARBs ...) these drugs can increase or decrease glomerular filtration. In Muskiet et al Nat Rev Nephrol 2014 we can read that the gut has been suggested to directly detect changes in the levels of ingested electrolytes and couple these changes to release of hormones and/or activation of neural pathways that regulate renal tubular and gastrointestinal transport.GLP-1 can be one of gut hormones and peptides implicated in gut-renal axis. The mechanism by which GLP-1 contributes to the increased excretion of sodium and solutes after ingestion is, in part, by increasing blood pressure and vasodilation of the afferent arteriole (through nitric oxide), increasing the renal blood flow and glomerular filtration, producing pressure natriuresis. Even, an impaired gut–renal axis in urinary sodium excretion might contribute to salt-sensitive hypertension. Although the natriuretic effect and the effect on glomerular filtration are adequately explained in the text, it is not related to the gut-renal axis.

Answer: As recommended, to better understanding for the audience of the action of GLP1 and GLP1RAs on kidney pathophysiology we have included a new section with a detailed explanation of the gut-renal axis mechanisms.

 3)In my opinion, one of the most interesting results of AWARD-7 is that at 52 weeks, eGFR was higher with dulaglutide 1∙5 mg (Chronic Kidney Disease Epidemiology Collaboration equation by cystatin C geometric LSM 34∙0 mL/min per 1∙73 m² [SE 0∙7]; p=0∙005 vs insulin glargine) and dulaglutide 0∙75 mg (33∙8 mL/min per 1∙73 m² [0∙7]; p=0∙009 vs insulin glargine) than with insulin glargine (31∙3 mL/min per 1∙73 m²). Dulaglutide treatment was associated with a significantly smaller decline in eGFR compared with insulin glargine over 52 weeks. These data suggest that dulaglutide could have specific therapeutic benefits that might slow progression of moderate-to-severe chronic kidney disease in type 2 diabetes. The association between dulaglutide treatment and reduced eGFR decline was most evident in participants with macroalbuminuria. Is the only study with GLP-1RA in which there was a reduced decline in eGFR. Although dulaglutide treatment was associated with a reduced decline in eGFR, the study was not designed to assess clinical endpoints because of the short 1-year treatment duration. Differences in rates of chronic kidney disease progression or end-stage renal disease will require long-term studies to substantiate that the eGFR benefit of GLP-1RA treatment translates to reduced rates of these events. (Tuttle KR, Lakshmanan MC, Rayner B, Busch RS, Zimmermann AG, Woodward DB, et al. Dulaglutide versus insulin glargine in patients with type 2 diabetes and moderate-to-severe chronic kidney disease (AWARD-7): a multicentre, open-label, randomised trial. Lancet Diabetes Endocrinol [Internet]. 2018 Aug 1 [cited 2018 Oct 13];6(8):605–17)

Answer: We extended these section with the results obtained in AWARD-7 with dulaglutide and changes in GFR, as suggested.

4)Another interesting question that in my opinion should be mentioned is that these drugs can be used up to a GFR of 15 mL/min/1.73 m2 (Górriz, J.L.; Soler, M.J.; Navarro-González, J.F.; García-Carro, C.; Puchades, M.J.; D’Marco, L.; Martínez Castelao, A.; Fernández- Fernández, B.; Ortiz, A.; Górriz-Zambrano, C.; et al. GLP-1 Receptor Agonists and Diabetic Kidney Disease: A Call of Attention to Nephrologists. J. Clin. Med. 2020, 9, 947). This fact implies the importance of this type of hypoglycemic agents for the nephrological community and DKD patients.

Answer: Thank you for your suggestion, in line 65 to 70 we have included a mini-resume of the safety of some GLP1-RAs in DKD.

We hope that we have properly addressed all the issues and you find this reviewed manuscript suitable for publication in the Journal of Clinical Medicine.

Yours sincerely,

Andreea Ciudin MD, PhD

Endocrinology and Nutrition Department
Morbid Obesity Unit Coordinator-EASO acredited Center of Excellence-COM
Hospital Universitari Vall d´Hebron
Universitat Autònoma de BarcelonaPg Vall d´Hebron 119-129, 08035, Barcelona
0034932744736/0034932746591e-mail: aciudin@vhebron.net

Round 2

Reviewer 1 Report

Dear Authors,

Thank you kindly for your responses and the additional work you have put in to this manuscript. Your corrections and updates have improved the flow and clarity, and also strengthened the overall position of your review. The additional paragraphs are nicely detailed and interesting to read. (Thank you also for picking up my typo).

I note, however, a couple of very minor things to update in the final manuscript.  The clinical trial numbers are missing from the revised manuscript as addressed at point 13. Also, I suggest the order of Table 1 should follow the same as the order of the paragraphs in section 4 for consistency. 

Regardless, I commend your effort in your revisions.

Author Response

REVIEWER 1-round 2

Dear Authors,

Thank you kindly for your responses and the additional work you have put in to this manuscript. Your corrections and updates have improved the flow and clarity, and also strengthened the overall position of your review. The additional paragraphs are nicely detailed and interesting to read. (Thank you also for picking up my typo).

I note, however, a couple of very minor things to update in the final manuscript.  The clinical trial numbers are missing from the revised manuscript as addressed at point 13. Also, I suggest the order of Table 1 should follow the same as the order of the paragraphs in section 4 for consistency. 

Regardless, I commend your effort in your revisions."

Answer: Thank you for your kind answer and for accepting the changes for the first round of review and for the pertinent observation in the second round. All your comments have significantly increased the quality of our paper.

We have added the NCT in Table 1 for each one of the clinical trials and also ordered the detailed explanation for each clinical trial in the text, following the same order as is Table 1.

We hope that we have properly addressed all the issues and you find this reviewed manuscript suitable for publication in the Journal of Clinical Medicine.

Yours sincerely,

Andreea Ciudin MD, PhD

Endocrinology and Nutrition Department
Morbid Obesity Unit Coordinator-EASO acredited Center of Excellence-COM
Hospital Universitari Vall d´Hebron
Universitat Autònoma de BarcelonaPg Vall d´Hebron 119-129, 08035, Barcelona
0034932744736/0034932746591e-mail: aciudin@vhebron.net
